# Whole-brain tissue mapping toolkit using large-scale highly multiplexed immunofluorescence imaging and deep neural networks

Dragan Maric [1,3,4✉], Jahandar Jahanipour [1,2,3], Xiaoyang Rebecca Li[2], Aditi Singh [2], Aryan Mobiny[2], Hien Van Nguyen[2], Andrea Sedlock[1], Kedar Grama [2] & Badrinath Roysam [2,4✉]

Mapping biological processes in brain tissues requires piecing together numerous histological observations of multiple tissue samples. We present a direct method that generates readouts for a comprehensive panel of biomarkers from serial whole-brain slices, characterizing all major brain cell types, at scales ranging from subcellular compartments, individual cells, local multi-cellular niches, to whole-brain regions from each slice. We use iterative cycles of optimized 10-plex immunostaining with 10-color epifluorescence imaging to accumulate highly enriched image datasets from individual whole-brain slices, from which seamless signal-corrected mosaics are reconstructed. Specific fluorescent signals of interest are isolated computationally, rejecting autofluorescence, imaging noise, cross-channel bleed-through, and cross-labeling. Reliable large-scale cell detection and segmentation are achieved using deep neural networks. Cell phenotyping is performed by analyzing unique biomarker combinations over appropriate subcellular compartments. This approach can accelerate pre-clinical drug evaluation and system-level brain histology studies by simultaneously profiling multiple biological processes in their native anatomical context.

[1] National Institute of Neurological Disorders and Stroke, Bethesda, MD 20892, USA. [2] Cullen College of Engineering, University of Houston, Houston, TX 77204, USA. [3] These authors contributed equally: Dragan Maric, Jahandar Jahanipour. [4] These authors jointly supervised this work: Dragan Maric, Badrinath Roysam. ✉email: maricd@ninds.nih.gov; broysam@uh.edu

Mammalian brain cytoarchitecture is a complex assemblage of multiple cell types, supported by an intricate microvascular network. Each cell has a molecular signature and morphology that defines its type (e.g., neuron, astrocyte, microglia, oligodendrocyte, endothelial cell, etc.), subtype (e.g., myelinating or non-myelinating oligodendrocyte), and functional state (e.g., resting, reactive, proliferating, apoptotic, phagocytic, etc.)[1–6]. These details are vital for understanding healthy and pathophysiological biological processes in the brain. Long-range brain connectivity implies that these processes can be spread across distant regions, requiring large-scale imaging. Unfortunately, conventional immunohistochemical (IHC) tissue imaging typically using only 3–5 molecular biomarkers of interest, and limited fields of view[7] does not capture this complexity. Additionally, human visual scoring of acquired imaging data is typically subjective, and limited to the visually detectable patterns[8], thus potentially missing latent cellular states that can only be revealed sensitively and specifically by a differential presence/absence/distribution using unique combinations of molecular biomarkers. Given these limitations, mapping complex processes in brain tissue requires investigators to piece together information painstakingly from a large number of bioassays and tissue samples, often from multiple tissue sources.

There is a compelling need for comprehensively and quantitatively profiling all relevant cellular and molecular players spread across all brain cell types and regions, in their natural microenvironmental niches and spatio-anatomical context[9]. This requires the ability to record high-resolution images of brain tissue covering a comprehensive panel of molecular biomarkers, over a large spatial extent, e.g., whole-brain slices, and automated ability to generate quantitative readouts of biomarker expression for all cells (including appropriate sub-cellular compartments), identifying the type/sub-type and phenotypic state of each cell based on unique combinations of biomarkers, and aggregating the resulting data at scales ranging from subcellular compartments, individual cells, local multi-cellular niches, to whole-brain regions.

In this regard, multiplex biomarker immunohistology on whole-brain tissue slices combined with high-resolution, large-scale epifluorescence microscopy offers detection of molecular specificity, structure preservation, and sufficient spatial resolution for discerning cellular morphology and protein localization within the correct anatomical context. Current alternatives to fluorescence microscopy come up short. Microarrays[10] provide a detailed molecular signature but entail a cytoarchitectural disruption and loss of cell-spatial information. Single-cell RNA sequencing provides a detailed transcriptomic profile but is not scalable to millions/billions of brain cells without losing spatial location. Mass spectrometry[11] offers a detailed peptide signature, but is expensive, less accessible, and provides low spatial resolution (~20 μm) compared to optical microscopy. However, fluorescence microscopy imaging is not without challenges when attempted on a large scale. It requires validated immunolabeling protocols that are capable of imaging large panels of molecular biomarkers across large swathes of brain tissue, and automated image scoring methods that can cope with staining variability, illumination non-uniformity, autofluorescence, background noise, photobleaching, spectral bleed-through, cross-labeling, stage movement errors, random artifacts, and tissue imperfections (e.g., inadvertent tissue folds/tears due to tissue slicing).

To address the above challenges, we present a comprehensive toolkit for achieving a major scaling of the multiplexing level and spatial extent (whole-brain slices) using a conventional epifluorescence microscope optimized for high-content imaging to phenotype all major cell classes resident to the whole brain, efficiently overcoming the fluorescence signal limitations, and achieving highly enriched and high-quality source imagery for reliable automated scoring at scale. Our goal is to accelerate system-level studies of normal and pathological brains, and pre-clinical drug studies by enabling targeted and off-target drug effects to be profiled simultaneously, in context, at the cellular scale. Our toolkit includes 3 major components: (1) optimized multiplex IHC staining and multispectral epifluorescence imaging protocols and associated computational tools; (2) robust fluorescence signal isolation algorithms; and (3) comprehensive deep learning-based methods for automated cell phenotyping at scale (Fig. 1a). This toolkit generates a comprehensive data table that can be analyzed without limitations and profiled at multiple scales

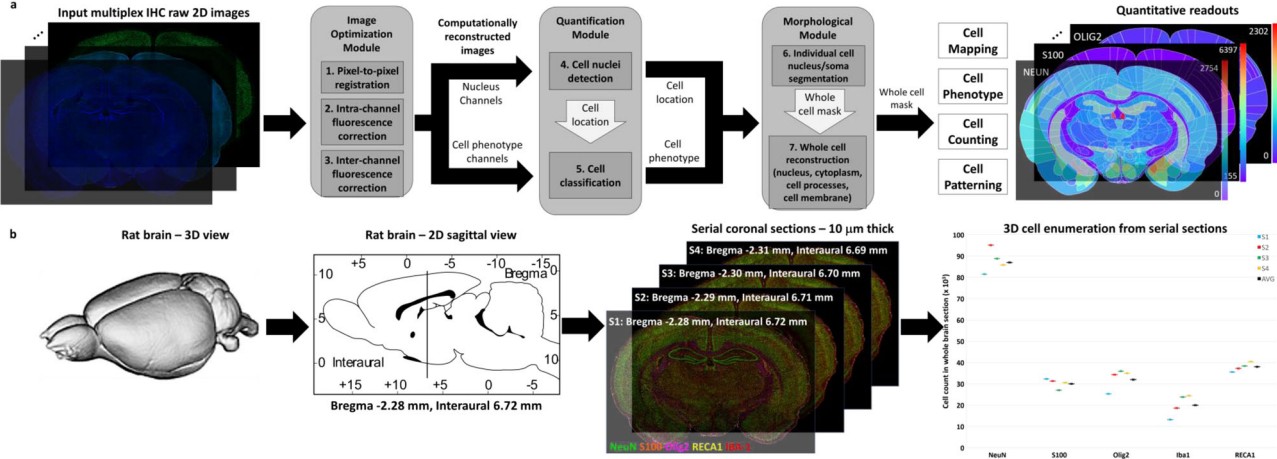

**Fig. 1 Overview of the whole brain tissue phenotyping pipeline for processing highly multiplexed immunohistological (MP-IHC) image datasets acquired using a customized wide-field multispectral epifluorescence imaging platform (refer to Fig. 2 for details) in conjunction with integrated open source computational modules for image reconstruction, optimization and quantitative deep machine learning data analyses (refer to Figs. 3–6 for details). a** The pipeline input consists of raw high-content 2D multispectral MP-IHC imaging datasets sourced from batch slide scans of 10 μm thick serial rat brain tissue sections iteratively probed with a myriad of biomarkers, which are then computationally processed for image registration, and intra- and inter-channel correction prior to deep learning-based quantitative analyses for cell phenotyping, cell counting, and anatomical mapping. **b** Additional modules include 3D reconstruction from MP-IHC image datasets sourced from multiple batches of serial whole rat brain tissue sections imaged in 2D and processed for volumetric 3D data analyses.

ranging from individual cells and multi-cellular niches to whole-brain anatomic regions in thin sections which enables high fidelity protein expression screening and data mining due to accessibility of target proteins to applied antibodies. The readouts can also be stacked to assess 3D brain immunohistology datasets from serial 2D sections (Fig. 1b), allowing comprehensive system-level studies of brain structure and function.

## Results

**Multiplex IHC staining, fluorescence imaging, and image reconstruction**. In the interest of widespread adoption, we developed our multiplex IHC staining and multispectral wide-field epifluorescence imaging strategy, as detailed in Methods section by adapting the use of standard laboratory protocols and conventional imaging equipment. With regards to the latter, we used a Zeiss AxioImager.Z2 scanning epifluorescence microscope with a standard motorized 10-position filter turret, a ×20, 0.8 NA Plan-Apochromat (Phase-2) non-immersion objective (Carl Zeiss), a 16-bit ORCA-Flash 4.0 sCMOS digital camera (Hamamatsu Photonics, Japan), and a 200 W X-Cite 200DC broad-spectrum excitation light source (Lumen Dynamics). In order to enable larger antibody panels, we performed 10-plex imaging using 10 custom self-contained excitation/dichroic/emission filter sets optimized for iteratively imaging up to 10 different spectrally compatible fluorescently tagged biomarkers at a time, over multiple iterative rounds of antibody staining and imaging (Fig. 2). This enables efficient fluorescence signal detection spanning the ultraviolet to near-infrared (350–800 nm) light spectrum, in conjunction with a comprehensive palette of validated primary antibody probes (Supplementary Table 1) and spectrally compatible fluorophore-conjugated secondary antibody probes (Supplementary Table 2). The signals of each fluorophore are resolved using excitation/dichroic/emission filter sets selected to minimize spectral crosstalk (Fig. 2a, Supplementary Tables 3 and 4). Iterative rounds of antibody staining and imaging are performed on the same tissue specimens, with carefully chosen combinations of biomarker probes, and an antibody stripping step added between each 10-plex biomarker screening round. Images are computationally aligned laterally and across staining rounds to reconstruct multiplex mosaic images of entire brain slices. For this, we developed an automated computational pipeline using Python and MATLAB® programming languages for combining the series of $N \times T \times R$ raw images collected as described above, where $N$ is the multiplexing level (typ. 10), $T$ the number of image fields (typ. 330), and $R$ is the number of repeated staining rounds (typ. 1–10), into a single multiplex mosaic image that contains only the corrected fluorescence signals of interest (Fig. 3). The raw images are stitched using the Zeiss microscope's software (Zen) and registered to correct for stage alignment errors by computing an affine spatial transformation for pixel-to-pixel registration (Fig. 3a). Then, signal corrections are performed, providing seamless 10–100 plex mosaics of whole-brain slices that provide sufficient spatial extent and spatial resolution to reveal cellular structures[12,13]. Importantly, they are suitable for automated scoring, for identifying a rich diversity of cell types (Fig. 2b, Supplementary Tables 5 and 6), functional states in situ, and spatial profiling of cellular distributions for comprehensive cytoarchitectural mapping in the context of brain anatomy (Fig. 1a).

**Computational signal isolation**. Isolating the specific fluorescence signal of interest from diverse potential sources of non-specific signals in acquired fluorescence images using direct physics-based methods is a forbidding task given the large number of parameters, most of which are variable or unknown,

and high computational requirements. The specific fluorescence signals arising from the immunolabeled targets may be corrupted by signal degradations including autofluorescence, photobleaching, sensor noise, non-uniform illumination, spectral mixing, cross-labeling, and microscope stage positioning errors (within and between staining and imaging rounds). We overcome this barrier using the following strategy. First, we make the well-founded assumption that the labeling and imaging protocols are sufficiently optimized to ensure that the specific fluorescent signals of interest are always brighter than the non-specific background. Second, we assume that the structures of interest have a local morphology that is distinct from the background. Under these assumptions, we use alternating sequential filters (ASF) that only require the minimum and maximum spatial scales $\sigma_{min}$ and $\sigma_{max}$ of the cellular objects to be specified, to identify and subtract the non-specific intra-channel signals including non-uniform illumination, spatially varying photobleaching[14], imaging noise[15] and tissue autofluorescence (Fig. 3b). Next, we developed an efficient semi-supervised sparse linear spectral unmixing algorithm to correct for spectral bleed-through[16], and cross-labeling[17] (Fig. 3c) as detailed in Methods section.

**Deep learning-based multiplex quantification**. Reliable detection of cell nuclei is a fundamental first step to automated image scoring. However, when imaging extended brain regions, we found that common DNA binding dyes (e.g., DAPI) exhibit so much staining variability that many cell nuclei are nearly undetectable (Fig. 4). We found that pan-histone markers provide a valuable complement to label such nuclei. For this reason, we trained deep neural networks to achieve reliable cell detection by co-analyzing images containing DAPI and pan-histone labeling markers (Fig. 4a–e). We developed a transfer learning-based approach to generate sufficient training samples to train the Faster-RCNN neural network[18] for detecting cell nuclei (Fig. 4f–j). This network generates a set of bounding boxes indicating detected cell nuclei (Supplementary Table 7). The bounding boxes are then used to classify major brain cell types (neurons, astrocytes, oligodendrocytes, endothelial cells and microglia) based on specific cell phenotype markers using an imporved Capsule Network[19,20] (Fig. 5a). Next, we developed algorithms to delineate key sub-cellular compartments (nucleus, soma, membrane, and processes), as illustrated in Fig. 6a–f. These compartments (masks) are used to generate localized measurements of biomarker expression for cell phenotyping analysis. These results are consolidated into a unified table of measurements (Supplementary Table 7), in which each row corresponds to a single cell, and the columns list the quantitative measurements for each cell. This table can be analyzed in a variety of ways, as illustrated by the following examples.

The first example shows the use of Hinton's Capsule Neural Network to identify cell types based on cell-type markers (Fig. 5a)[19,20]. Capsule Networks analyze multiple cellular features jointly by encapsulating them into a vector, whose lengths that can be thresholded for reliable cell classification (Fig. 5b, c) and phenotyping (Supplementary Table 7). This method overcomes the limitations of traditional phenotyping of histogram of intensity of the biomarker by using a set of comprehensive features (Fig. 5d–h) and corrects the mis-identified cells. The second example (Fig. 6h–m) shows the ability to identify a rich set of neuronal sub-types (glutamatergic, GABAergic, cholinergic, and catecholaminergic) based on combinations of molecular markers (GLUT, GAD67, CHAT, and TH). The functional status (proliferation and apoptosis) of each cell is also derivable from the related markers (PCNA and CC3) using Supplementary Table 7. The third example (Fig. 6b–f) illustrates the analysis of molecular markers in cells with arbors to infer cell states based on sub-cellular

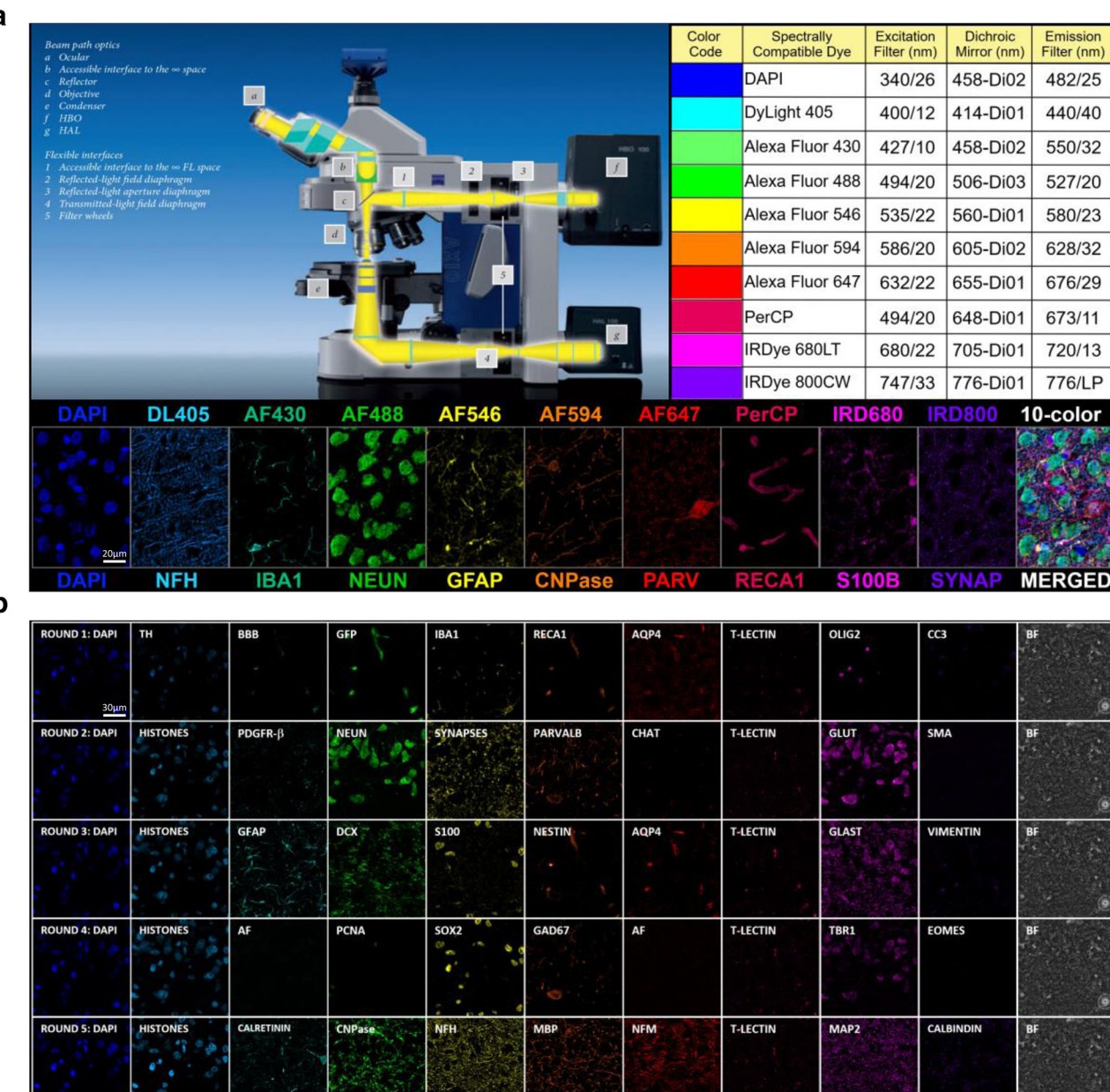

**Fig. 2 Overview of the multiplex IHC staining and multispectral epifluorescence imaging platform. a** The imaging platform optimally utilizes the commercial availability of a wide selection of spectrally compatible fluorophores across the full usable light spectrum ranging from ultraviolet to near-infrared (350–800 nm). Components of the wide-field epifluorescence microscope required for 10-color multispectral imaging include a Zeiss AxioImager. Z2 upright fluorescence microscope (left) equipped with a high-resolution objective, a high sensitivity digital camera, a broad-spectrum light excitation source and 10 self-contained excitation/dichroic/emission filter sets optimized to detect up to 10 commonly used or custom-derived spectrally compatible fluorescent reporters (Supplementary Table 2), with minimal spectral cross-talk (Supplementary Table 4), as specified in the excitation/dichroic/emission filter table (right), and exemplified in the included representative images (bottom) of major brain cell types as visualized using a 10-plex immunostaining protocol outlined in Supplementary Table 5. **b** The basic 10-color fluorescence imaging platform can be user-configured to include different combinations of customized filters to accommodate optimal imaging of different selections of commonly applied and custom-derived fluorescent dyes (Supplementary Table 2) and used iteratively with different selection of immunocompatible biomarkers to label a myriad of brain cell types exemplified in the included representative images in the 50-plex image dataset as visualized using a 5-round immunostaining protocol outlined in Supplementary Table 6.

localization of the markers. Our analysis is limited to the basal portions of the arbors to the extent captured by the 10 μm tissue thickness used to slice brain tissues for immunohistology. In this example, astrocytes were reconstructed using Sox2 (nuclear mask), S100β (soma mask), and GFAP (arbor processes mask) channels. IBA1 channel was used to reconstruct microglia soma and processes. MAP2 and NeuN channels were used for basal neuron arbor reconstruction. Olig2 (nuclear mask) and CNPase (soma and

processes mask) were used to reconstruct oligodendrocytes. This analysis produces a comprehensive table in which each entry corresponds to an identified cell (along with its spatial, morphological, and molecular measurements over cell compartments, and determinations of cell type and cell functional status), with negligible loss (<5%), as summarized in Fig. 6g and Supplementary Table 7. These tabular data can readily be profiled at multiple scales including whole-brain anatomical mapping (Fig. 6n–p) and analyzed using standard

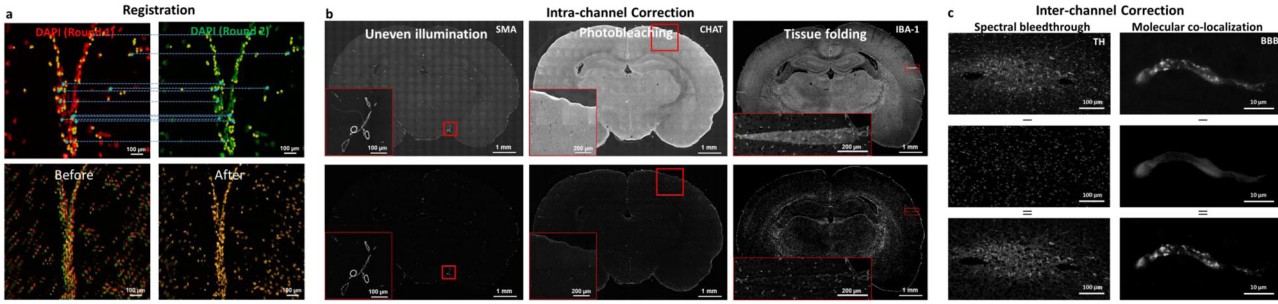

**Fig. 3 Overview of the post-acquisition image optimization module. a** Pixel-to-pixel registration of the separate imaging rounds by detecting the nuclei landmarks from the DAPI channel in each round (top) and applying affine transformation to register the images to the target round (bottom). **b** Intra-channel correction of non-specific signals introduced during imaging including uneven illumination (left), photobleaching (center), and tissue folds (right) using Alternating Sequential Filters (ASF). The registered image (top) is subtracted from the estimated intra-channel non-specific signal to get the corrected signal (bottom). **c** Inter-channel correction of non-specific signal to extract spectral bleed-through (left) and molecular co-localization (right) utilizing a semi-supervised sparse linear spectral unmixing approach. Original signal (top) is subtracted from the estimated mixed channel (middle) to get the specific fluorescent signal of interest (bottom).

statistical packages. The spatial relationships of the quantified cells can be grouped based on the anatomical region they are located. We can leverage the anatomical position information to study the differences in the population of cells in different anatomical regions or between the same regions in different sections or different animals.

Finally, we expanded the analysis of the proposed pipeline (Fig. 1a) for associating 3D brain immunohistology datasets by comparing the 2D phenotyping results from the immediately adjacent serial whole-brain sections to assess a volumetric dataset in the context of 3D brain anatomical mapping (e.g., Fig. 1b). Our aim is not to present a method for true 3D reconstruction representing the connectome of brain cytoarchitecture but to expand the 2D immunohistology results from individual whole-brain slices by associating the cellular phenotyping across adjacent thin serial sections. This representation intends to fill in the gaps depicted in the published atlases which illustrates the consistency of the cellular composition in the adjacent sections. The effectiveness of our deep learning-based analysis provides an additional tool for rigorous scalable validation of future 3D analyses[21].

## Discussion

By synergistically integrating iterative high-content 10-plex IHC immunostaining with 10-color fluorescence imaging using conventional laboratory techniques and instrumentation, together with open source computational pipeline for specific signal isolation, and robust automated signal scoring using deep learning methods at scale, we developed a versatile toolkit that can be readily implemented for widespread use with the potential to transform 2D and 3D brain histology studies requiring comprehensive cellular profiling from single and serial slices of brain tissue.

Compared to existing multiplex IHC biomarker screening techniques using iterative immunostaining and computational analysis, that typically build from low-plex panels of directly conjugated antibodies (3–4 biomarkers screened per cycle) using small tissue samples and/or limited fields of view and are mostly focused on tumor biology[22], our methods are more flexible, scalable and efficient, enabling multiplex IHC imaging and computational analysis of up to 10 different biomarkers of interest at the same time using direct or indirect IHC immunostaining protocols. The prior work of Micheva et al.[23] imaged brain tissue using array tomography on very thin (49 nm) sections, but screened a limited number of biomarkers compared to the present work, predominantly focused on neuronal

phenotyping in smaller ~1 mm regions of the brain. Our work represents a different tradeoff, with a lower spatial resolution, but greater coverage in terms of molecular markers and spatial extent, and with an emphasis on routine usage in systems studies. Our protocols can be expanded using iterative staining and imaging cycles to screen a potentially unlimited number of biomarkers of interest per individual tissue samples, with a "clean slate" enabled on individual tissue specimens with each probing cycle using antibody stripping steps to remove tissue-bound antibodies between each antibody staining round. The latter approach has a significant advantage over currently used multiplex IHC methods that rely on photobleaching and/or fluorophore quenching to remove residual fluorescence signals on tissue bound antibodies between iterative staining and imaging rounds[22], instead of stripping those antibodies off tissue samples, leading to possible steric hindrance and antibody/antigen binding issues with each subsequent antibody probing cycle on the same tissue samples. Our computational analysis pipeline is inherently scalable, both in terms of molecular signature as well as spatial coverage. Our core approach is also adaptable to other tissues. This can accelerate systems-oriented studies by providing quantitative profiles of all the molecular and cellular players at once, in their detailed spatial context. It can be used as part of an assay to reveal both the targeted as well as off-target side effects of drug candidates at once, reducing the animal usage and eliminating the need to piece together a process narrative from a series of low-content observations from multiple animals. The core methods presented here can form the basis for advanced turnkey quantitative neurohistology systems including antibody labeling kits that replace conventional H&E-based methods. Although the cost of multiplex IHC processing of individual and serial tissue slices is higher compared to conventional low-plex methods, our approach can be cheaper overall since less tissue needs to be processed to definitively answer specific questions using high-content (>10-plex) targeted biomarker screening and the analyses of the results are directly comparable on the same cells and cell populations of interest rather than correlating the fragmented low-content (< 5-plex) results sourced from multiple sections.

Our immunostaining protocols can be scaled up/down as needed and applied to successive serial sections for whole-brain 3D reconstruction. Even a single 10-plex round of staining is quite informative. More extensive molecular signatures can be acquired by performing additional rounds of staining and imaging, or vice versa, without compromising tissue integrity. Our computational methods rely on streaming computer architectures

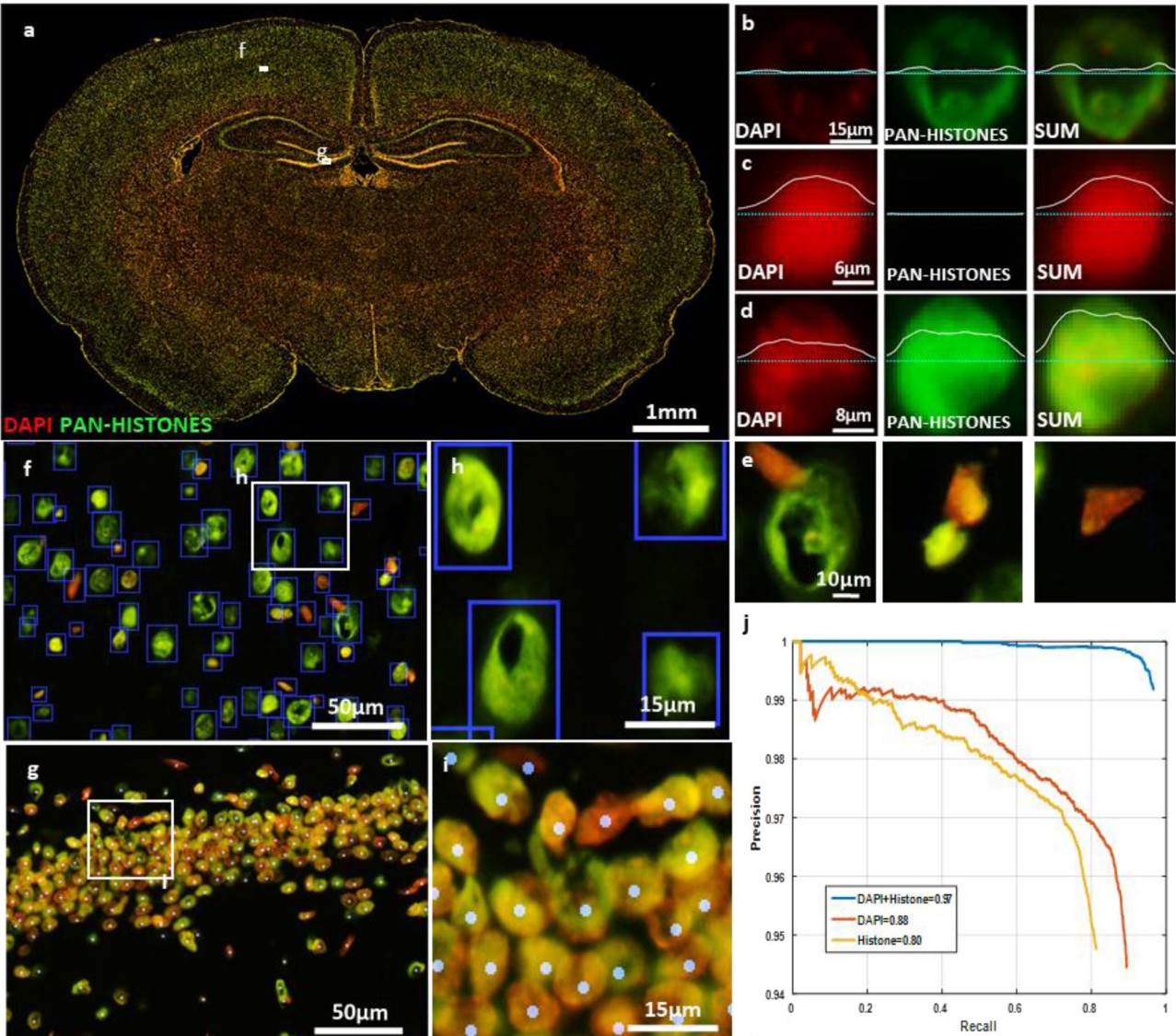

**Fig. 4 Improved nuclei detection in whole brain images using a multiplexed approach. a** Montage showing variability in DAPI (pan-nuclear DNA stain) and histone (pan-nuclear protein) expression labeling across the whole rat brain tissue slice. **b–e** The sum of the DAPI and pan-histone markers capture nuclear morphology more reliably than either marker alone. **f–i** Close-up images showing DAPI and pan-histone labeling variations in a sparse and a densely packed region and the generated location (bounding box) results of the proposed model for reliable detection of the cell nuclei using transfer learning approach in conjunction with a Faster-RCNN network. **j** Receiver Operating Characteristic (ROC) curve of cell nuclei detection using single marker and combination of markers shows significant improvement in performance when both markers are included.

(GPU arrays), making them scalable to larger image datasets, e.g., whole primate and human brains and other large organ tissues, without limit.

To promote broad adoption, our method is designed around a conventional epifluorescence microscope with commercially available components (objectives, digital camera, filter sets, broad-spectrum light source, and a computer-controlled stage), and the computations are handled by GPU-equipped computers. For practicality and efficiency, we focused on thin (10 μm) slices that are imaged two-dimensionally to avoid antibody/dye penetration limitations and complexities associated with 3D imaging and automated image analysis[24]. The proposed computational pipeline is specifically designed to generate 2D immunohistology results from individual whole-brain slices which is expandable for 3D segmentation[25] and reconstruction[26] to better understand the connectome of brain cytoarchitecture at cellular, niche, and organ levels[27]. Cellular measurements are exported to flow cytometry

standard (FCS) and Image Cytometry Experiment (ICE) file formats for visualization and statistical profiling using common commercially available software tools (e.g., FCS Express, De Novo Software; FlowJo, BD Biosciences, Kaluza, Beckman Coulter, etc.). We provide sample images, staining and imaging protocols, and open source code written in Python and MATLAB® programming languages[28]. The images and the results could be visualized in commercially and open-source available software tools (e.g., Adobe Photoshop, ImageJ, Napari[29], etc.). The cellular measurements can be also studied in relation to the anatomical mapping which we manually fitted in this study as a proof of principle. However, this approach can certainly be automated using existing computational methods from published atlases[30].

## Methods

**Specimen preparation.** We used an ex vivo rat brain model as proof-of-concept for our multiplex IHC staining and imaging method. All animal handling

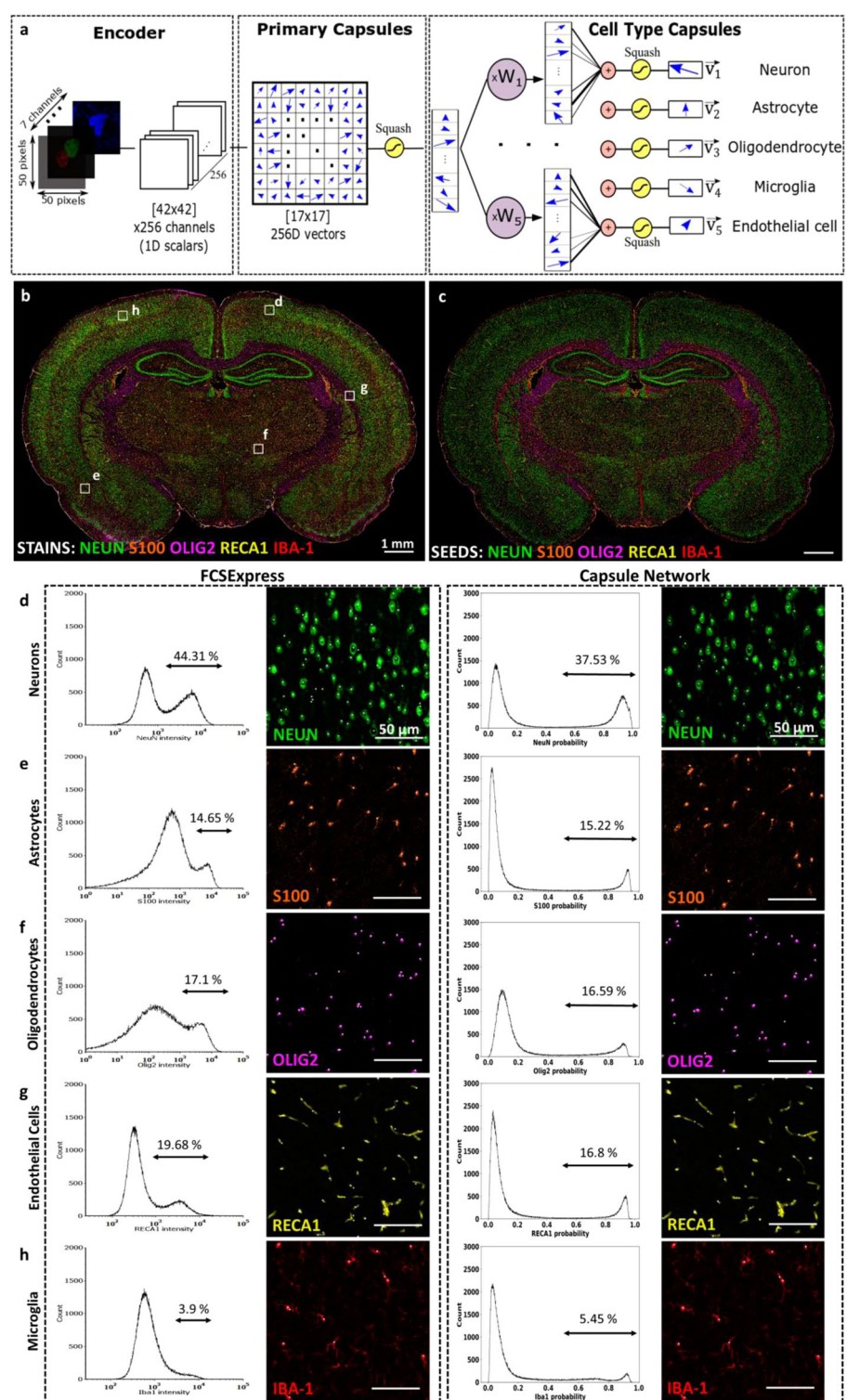

**Fig. 5 Improved methods for cell-type classification and quantification using deep morphological features compared to traditional intensity levels thresholding. a** Architecture of the proposed Capsule Network (CapsNet) to extract more comprehensive features for more accurate cell classification. **b** Pseudo-colored multichannel montage image of 5 major brain cell types. **c** Computational reconstruction of major cell-type montage from classified seeds (centers of bounding boxes) pseudo-colored to match the color of the actual biomarker expression recapitulating the specific cellular distribution of each cell type in the original raw image in (**b**) with high fidelity. Histogram of lengths of capsule vectors in the last layer of the network (right) compared to traditional phenotyping approach by thresholding on the histogram of mean signal intensity of the major brain cell-type classification biomarkers (left) measured inside individual bounding boxes using DAPI+histone for seed detection (depicted in Fig. 4f-i) identifying (**d**) neurons (NeuN), (**e**) astrocytes (S100β), (**f**) oligodendrocytes (Olig2), (**g**) endothelial cells (RECA1), and (**h**) microglia (Iba1). The histograms of the proposed method shows bimodal distribution with well-separated peaks for better separation of negative and positive population of cells for each cell phenotype with enlarged regions of interest from insets with single biomarker channels and overlaid classified seeds (white dots) illustrates the complete match of the generated cell phenotypes in the raw images.

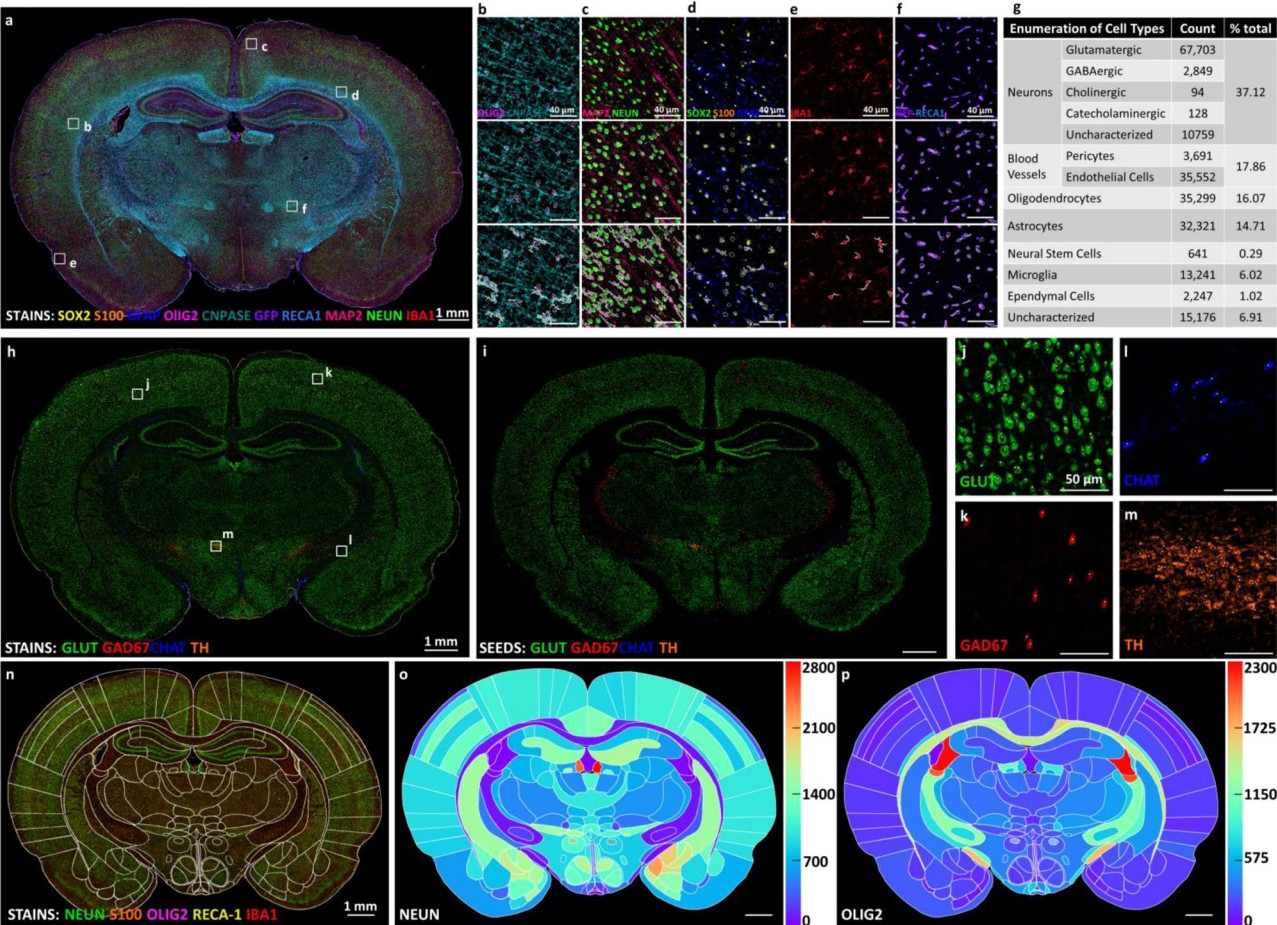

| Enumeration of Cell Types | | Count | % total |
|---|---|---|---|
| Neurons | Glutamatergic | 67,703 | 37.12 |
| | GABAergic | 2,849 | |
| | Cholinergic | 94 | |
| | Catecholaminergic | 128 | |
| | Uncharacterized | 10759 | |
| Blood Vessels | Pericytes | 3,691 | 17.86 |
| | Endothelial Cells | 35,552 | |
| Oligodendrocytes | | 35,299 | 16.07 |
| Astrocytes | | 32,321 | 14.71 |
| Neural Stem Cells | | 641 | 0.29 |
| Microglia | | 13,241 | 6.02 |
| Ependymal Cells | | 2,247 | 1.02 |
| Uncharacterized | | 15,176 | 6.91 |

**Fig. 6 Summary of quantification and morphological modules. a** Pseudo-colored multichannel montage of the whole rat brain image scan (left) showing biomarkers used for morphological masking. **b–f** Selected close-up regions illustrate the original image (top), nucleus/soma mask (middle) and whole cell mask (bottom) including nucleus, cytoplasm, cell processes, plasma membrane for each major brain cell type including oligodendrocytes (**b**), neurons (**c**), astrocytes (**d**), microglia (**e**) and endothelial cells (**f**). **g** Summarized table of enumeration and percentage of cell types and subtypes in a single whole-brain 2D tissue slice shown in (**a**). **h** Pseudo-colored multichannel montage image of neuronal cell-subtype classification biomarkers including glutamatergic (GLUT), GABAergic (GAD67), cholinergic (CHAT), catecholaminergic (TH), and uncharacterized neurons. **i** Computational reconstruction of neuronal cell-subtype montage from classified seeds pseudo-colored to match the color of the actual biomarker expression recapitulating the specific cellular distribution in the original raw image with high fidelity. **j–m** Enlarged regions of interest from insets in panel (**h**) with single biomarker channels and overlaid classified seeds (white dots) illustrates the complete match of the generated cell phenotypes in the raw images. **n** Manually-fitted Paxinos atlas on the 2D brain section to quantify the number of the cells per unit area for phenotypic analysis of cells in defined anatomical regions. Heatmap plot showing the number of positive cells per $10^6$ pixel$^2$ area, exemplified for neurons (**o**) and the oligodendrocyte (**p**) population.

procedures were approved by the Institutional Animal Care and Use Committee (IACUC) of the National Institute of Neurological Disorders and Stroke (NINDS, Bethesda, MD). Briefly, 8-week-old male Lewis LEW-Tg(CAG-EGFP)YsRrrc transgenic rats, initially sourced from Rat Resource and Research Center (Columbia, MO) and bred in-house at NINDS, were deeply anesthetized and transcardially perfused with heparin infused phosphate-buffered saline (PBS) pH 7.4 followed by 4% paraformaldehyde (PFA) in PBS. The brains were promptly removed from the skull and post-fixed in 4% PFA/PBS at 4 °C for 24 h, before sequentially undergoing cryoprotection in graded 10%, 20%, and 30% (w/v) sucrose solutions in PBS until the tissue specimens completely sank in each solution. The brains were then embedded in Optimum Cutting Temperature (OCT) medium, snap-frozen in isopentane and 10 μm thick whole-brain coronal sections cut in the areas immediately adjacent to coordinates interaural 6.72 mm and bregma −2.28 mm using the Rat Brain Atlas[31] as reference. These coordinates showcase a diverse anatomy of major brain regions, including the cerebral cortex, hippocampus, striatum, thalamus, and the hypothalamus. Cryosections were mounted on Leica Apex Superior Adhesive Slides (VWR, product# 3800080), which proved optimal for performing multi-round multiplex IHC staining protocols with minimal loss of tissue sections during repeated antibody stripping and heat-induced antigen retrieval steps detailed below.

**Multiplex IHC staining and imaging.** Current multi-round multiplex IHC bio-marker screening techniques using iterative immunostaining with low-plex panels

of directly conjugated antibodies (3-4 biomarkers screened per cycle) published to date[22] or offered by commercial automated systems (MACSima, Miltenyi Biotech; CODEX, Akoya Biosciences) have largely been optimized for tumor cell biology. For efficient high-order multiplex imaging of brain tissue, we developed an antibody staining protocol combining up to 10 well-characterized immunocompatible and individually validated primary antibodies of interest in a single staining cocktail mixture, including one mouse antibody from each available IgG subclass (IgG1, IgG2a, IgG2b, IgG3), plus one mouse or rat IgM antibody, plus one IgG antibody raised from each of the following non-mouse hosts (rat, hamster, rabbit, guinea pig, chicken, sheep, directly conjugated goat). All multiplexed antibody panels have been tested to stain their intended target antigens in the rat brain tissues and validated to show insignificant cross-reactivity and non-specific binding. Supplementary Table 1 exemplifies a growing list of empirically-tested multiplex IHC compatible primary antibodies to screen for virtually any and all different cell types either resident to the rat brain or infiltrating into the brain in response to diverse neuropathological conditions, with many of these antibodies also cross-reactive to their specific targets in mouse and human brain tissues. To visualize the unconjugated primary antibodies, we used highly cross-adsorbed secondary antibodies with each conjugated to one of the following spectrally compatible dyes for 10-color fluorescence slide scanning: DY-395XL, DyLight 405, Alexa Fluor 430, Alexa Fluor 488, Alexa Fluor 546, Alexa Fluor 594, Alexa Fluor 647, PerCP, IRDye 680LT, and IRDye 800CW (see Supplementary Table 2 for full listing of all commercially available off-the-shelf or custom-ordered spectrally compatible fluorophore-conjugated secondary antibodies optimized for 10-color

epifluorescence imaging). Furthermore, if using primary antibodies from the same host and immunoglobulin class or subclass, it is also possible to combine up 10 different antibodies with each directly conjugated to one of aforementioned spectrally-compatible fluorophores. Optimal detection of these fluorophores using widefield epifluorescence microscopy, as detailed below, also necessitated the customization of standard off-the-shelf filter sets typically used for imaging the aforementioned dyes (Supplementary Table 3) to a more stringent optical configuration of the exciter, dichroic, and emitter filters with more narrow bandpass and/or off-peak excitation/emission properties to minimize filter crosstalk to <10% bleed-through signal transmission as detailed in Supplementary Table 4.

The multi-round iterative multiplex IHC staining protocol is described as follows. Prior to the first round of immunostaining, the cryosections are thawed from −80 °C storage to room temperature (RT) and left to completely air-dry overnight to assure firm tissue bonding with the microscope slide. The entire immunostaining protocol is then carried out at RT. The sections are first permeabilized in graded 70%, 80%, 95% methanol (2 min/step) then rehydrated and washed in distilled water (dH$_2$O). Sections are then completely submerged in 10 mM Sodium Citrate buffer pH 6.0 and antigen unmasking is carried out with a 2-minute heat mediated antigen retrieval using an 800 W microwave with a turntable (GE model PEM31DFWW) set at 100% power. Sections are then sequentially treated with 15 min incubation at RT in neat FcR blocking solution (Innovex Biosciences, NB309) to block endogenous Fc receptors, followed by incubation in neat Background Buster solution (Innovex Biosciences, NB306) to block non-specific binding of exogenously applied primary and secondary antibodies. After the blocking steps, the sections are first immunoreacted using a cocktail mixture of unconjugated (or directly conjugated) immunocompatible primary antibodies (or ligands) of interest, with each primary antibody used at 1 μg/ml final concentration (i.e., 1:1,000 dilution of 1 mg/ml stock concentration), as exemplified in Supplementary Tables 5 and 6, with all staining reagents diluted in PBS supplemented with 1 mg/ml BSA, and incubated for 60 min at RT, then washed 3× in PBS and 3× in dH$_2$O (1 min/wash) prior to application of appropriate matching and spectrally compatible secondary antibodies as referenced above and in Supplementary Table 2, with each secondary antibody also used at 1 μg/ml final concentration (i.e., 1:1,000 dilution of 1 mg/ml stock concentration). Sections are then washed again 3× in PBS and 3× in dH$_2$O and subsequently counterstained with 1 μg/ml DAPI (Thermo-Fisher Scientific) to serve as a reference channel for autofocus during image scanning and pixel-to-pixel registration of separate image datasets sourced from iterative rescanning of the same specimen with different 10-plex biomarker panels. The slides with 10-color fluorophore-labeled tissue sections are then cover-slipped using Immu-Mount medium (Thermo-Fisher Scientific, MI). Sections are then imaged using an Axio Imager.Z2 10-channel scanning fluorescence microscope (Carl Zeiss, Thornwood, NY) equipped with a × 20, 0.8 NA Plan-Apochromat (Phase-2) non-immersion objective (Carl Zeiss), a 16-bit ORCA-Flash 4.0 sCMOS digital camera (Hamamatsu Photonics, Japan) sensitive to a broad-spectrum of emission wavelengths, including those approaching infrared, a 200 W X-Cite 200DC broad-spectrum light excitation source (Lumen Dynamics), and 10 self-contained excitation/dichroic/emission filter sets (Semrock, Rochester, NY) optimized to detect the following combination of fluorophores with minimal spectral crosstalk: Combination 1 - DAPI, DyLight 405, Alexa Fluor 430, Alexa Fluor 488, Alexa Fluor 546, Alexa Fluor 594, Alexa Fluor 647, PerCP, IRDye 680LT, and IRDye 800CW (see Supplementary Table 4 Custom 10-color Filter Setup 1 for filter specifications and spectral compatibility) or Combination 2 - DAPI, DyLight 405, DY395XL, Alexa Fluor 488, Alexa Fluor 546, Alexa Fluor 594, Alexa Fluor 647, PerCP, IRDye 680LT, and IRDye 800CW (see Supplementary Table 4 Custom 10-color Filter Setup 2 for filter specifications and spectral compatibility). Each labeling reaction was sequentially captured in a separate image channel using a filtered light through an appropriate fluorescence filter set and the image microscope field (600 × 600 μm) with 5% overlap, individually digitized at 16-bit resolution using the ZEN 2 image acquisition software (Carl Zeiss). For multiplex fluorescence image visualization, a distinct color table was applied to each image to either match its emission spectrum or to set a distinguishing color balance. The pseudo-colored images were then converted into 8-bit BigTIFF files, exported to Adobe Photoshop, and overlaid as individual layers to create multi-colored merged composites. For the computational image analyses detailed below, microscope filed images were seamlessly stitched in Zen 2 and exported as raw uncompressed 16-bit monochromatic BigTIFF image files for further image optimization and processing, as detailed below.

Subsequent rounds of tissue specimen re-staining and re-imaging involved the following iterative steps. After imaging the first round of 10-plex biomarkers, as exemplified in Fig. 2a (bottom panels) and Fig. 2b (first row panels), tissue bound primary and secondary antibodies (and fluorescently tagged ligands) were both stripped off the slides after a 5-minute incubation step at RT in NewBlot Nitro 5X Stripping buffer (Li-Cor Biosciences) followed by 1-minute additional heat mediated antigen retrieval step in 10 mM Sodium Citrate buffer pH 6.0 using the same 800 W microwave mentioned above set at 100% power. The above processing cycle, beginning with tissue re-blocking in FcR Blocking and Background Buster solutions, was then repeated and the same sections then incubated using a second mixture of primary antibodies and appropriate secondary antibodies, as exemplified in Supplementary Table 6 and Fig. 2b (second row panels). In the 5-round 50-plex IHC staining example focused in this study, the

whole process was sequentially repeated three more times using select first and second step reagents as listed in Supplementary Table 6 and exemplified in Fig. 2b (panels in rows 3–5).

**Mosaicing of image microscope fields.** Methods for stitching a series of image microscope fields to form a seamless mosaic image by correcting for stage translation errors are now well established[32], and incorporated into commercial microscope control software. Since our imaging was performed on a Zeiss system, we used the built-in Zen software (Zeiss) to align the image microscope fields laterally for each staining round.

**Registration of channels between staining rounds.** For aligning mosaics from one staining round to the next to single-pixel resolution, we noted that the tissue slices are minimally deformed by the processing at each staining round since they are mounted firmly on glass slides. An affine spatial transformation is adequate to describe these small deformations. For registration, we found that DAPI-stained cell nuclei provide reliable spatial landmarks[32]. For this reason, we include DAPI staining in each round. From the DAPI channel, we extracted key points and pattern descriptors using Oriented Fast and Rotate BREIF (ORB)[33]. These are more robust to imaging noise, and computationally more efficient compared to the Scale Invariant Feature Transform (SIFT)[34]. For matching pairs of key points, we used the Hamming distance metric. To estimate the affine spatial transformation between pairs of image mosaics, in a manner that is robust to outliers, we employed RANdom SAmple Consensus (RANSAC)[35]. To achieve scalable handling of mosaic images, we implemented RANSAC over cropped windows rather than the full mosaic to accelerate the keypoints extraction speed using parallel processing techniques. This approach is scalable, robust, and accurate to within a pixel across the entire tissue slice. Figure 3a shows a representative example of image matching using this procedure.

**Fluorescence signal correction.** The aligned fluorescence data are consolidated to produce a seamless 36-channel (42,906 × 29,286 pixels) raw mosaic containing only the biomarker signals of interest, correcting for non-uniform illumination, photobleaching[14], imaging noise[15], tissue autofluorescence, spectral bleed-through[16], and cross-labeling[17]. The raw fluorescence image $I^c(x,y)$ for channel c is modeled as the sum of the specific fluorophore-generated signal of interest, $I_s^c(x,y)$, background signal $I_B^c(x,y)$ comprising of the tissue autofluorescence that is modulated by the non-uniform illumination and photobleaching, and cross-channel bleed-through arising from spectral overlap, and molecular co-localization signals arising from non-specific labeling of one or more markers in other channels.

In order to extract the specific signal of interest without the challenges associated with a full-blown physics-based modeling, we developed the following strategy. We made the well-founded assumption that the immuno-fluorescent labeling and imaging protocols are optimized to ensure that the signal of interest is always brighter than the background signals. We also assumed that the structures of interest (cells and parts thereof) have a local morphology that is distinct from the background. With these assumptions, we developed a fast approach using alternating sequential filters (ASF)[36] that only requires the minimum and maximum scale factors $\sigma_{min}$ and $\sigma_{max}$ of the cellular objects to be specified. The combined background signal $I_B^c(x,y)$ is estimated as $ASF(I^c(x,y),\sigma_{min}, \sigma_{max})$, and subtracted from $I^c(x,y)$. Next, we developed a robust unmixing algorithm that corrects each channel for cross-channel bleed-through as well as molecular cross-labeling. We estimated the cross-channel bleed-through fractions $\alpha_{c'}$ using the Least Absolute Shrinkage and Selection Operator (LASSO)[37] by introducing an $l_1$ regularization term to the linear unmixing[38] as $\min_{\alpha_{c'}}\|I^c(x,y) - \sum_{c'=1}^{10} \alpha_{c'} I^{c'}(x,y)\|^2 + \|\gamma\alpha_{c'}\|_1$ where $\gamma$ is the trade-off constant of sparsity. It accounts for the fact that the cross-talk occurs over a sparse set of channels, providing a more reliable estimate compared to least-squares methods. To account for all inter-channel signals we proposed a semi-supervised linear unmixing algorithm by automatically estimating bleed-through channel fraction ($\alpha_{c'}$) and cross-labeling channel fraction from user ($\beta_{c''}$) as:

$$\min_{\alpha_{c'}, \beta_{c''}} \left\| I^c(x,y) - \sum_{\substack{c'=1 \\ c' \neq c''}}^{10} \alpha_{c'} I^{c'}(x,y) - \sum_{\substack{c''=1 \\ c'' \neq (c, c')}}^{10} \beta_{c''} I^{c''}(x,y) \right\|^2 \quad (1)$$

$$s.t. \begin{cases} 0 \leq \alpha_{c'} \leq e_{c'}, 0 \leq \beta_{c''} \leq e_{c''} \\ \sum_{\substack{c'=1 \\ c' \neq c''}}^{10} \alpha' + \sum_{\substack{c'=1 \\ c' \neq c''}}^{10} \beta_{c''} = K \\ e_{c'} \in \{0, 1\}, e_{c''} \in E \end{cases} \quad (2)$$

where $e_{c'}$ is the existence of bleed-through channels $c'$ (i.e., $e_{c'}$ is 1 for channels with bleed-through and 0 otherwise) derived automatically using constrained

LASSO[39], $e_{c''}$ is the existence of cross-labeling channels $c''$ (i.e., $e_{c''}$ is 1 for channels with cross-labeling and 0 otherwise) provided by the user from $E$, and $K$ is the maximum number of channels for inter-channel correction including both bleed-through and cross-labeling. Optimization algorithm will solve for $\alpha_{c'}$ and $\beta_{c''}$, and subtract the fraction of non-specific signal from the original channel. It has been empirically seen that $K=3$ will count for both bleed-through and molecular cross-labeling.

To optimize the inter-channel correction estimation, we use a Region Of Interest (ROI) selected by the user that contains background, auto-fluorescence, and fluorescence signal of interest. The unmixing algorithm will estimate the parameter based on the selected ROI which is faster than estimated on the whole image and enables the algorithm to work on desktop machines. Selecting the ROI also helps us addressing the imaging artifacts appearing in the background. Since the selected ROI is guaranteed to have the brightest fluorescence signal and it is debris-free, we can threshold any pixel that is brighter than the brightest pixel in the ROI to remove the random bright spots/debris.

To use this method, the algorithm works without the knowledge of bleed-through channels and the user merely needs to specify the pairs of channels for which cross-labeling is expected to occur (as vector $E$). All other parameters are estimated automatically. This procedure also copes with small tissue tears, folds, and sediment residues occasionally introduced during the staining process, as illustrated in Fig. 3b, c. The resulting images contain only the fluorescence signals of interest without the background and cross-channel interference and are ready for reliable automated image analysis.

**Reliable detection of cell nuclei in large multiplex images**. Detection of cell nuclei is a first step for cell profiling since all the cells of interest are nucleated. We found that the widely used DNA stain 4′,6-diamidino-2-phenylindole (DAPI) is inadequate for highlighting all cell nuclei across a brain slice. DAPI staining varies by brain region (Fig. 4a), and the staining can be incomplete or weak for many cells. To overcome this limitation, we added a pan-histone marker to complement DAPI. As shown in Fig. 4b–d, a summation of the DAPI and pan-histone signals reveals nuclei far more reliably compared to either stain alone. Even with this combined staining, detecting cell nuclei is challenging since they exhibit considerable morphological variability, and dense packing in some brain regions (Fig. 4e). Deep neural networks have the representational capacity to cope with these challenges. The challenge is to generate a sufficiently large corpus of training data. Fortunately, these networks are robust to occasional errors in the training data. With these factors in mind, we developed the following partly automated training strategy. First, we noted that conventional nuclear segmentation algorithms work well in most brain regions, but fare poorly on densely-packed cell ensembles (such as hippocampus). With this in mind, we trained a Faster RCNN[18] over ~200,000 nuclear segmentation results that were generated automatically by our earlier method[40,41] on the sum of DAPI and the pan-histone labels. The RCNN inputs consisted of the DAPI and pan-histone channels, and it was initialized with pre-trained weights from the ImageNet database[42]. The feature extraction module of this network was chosen to be the Inception ResNet V2[43]. In order to improve the performance of this network on the densely packed regions, we manually generated two corpuses of ~8,500 cells from 4,000 × 6,000 pixel region of the left and right hippocampal regions, for training and validation. To minimize the manual effort, we built these corpuses by manually editing the automatically generated cell detections for these regions. The Faster RCNN was then retrained on one of the corpuses by transfer learning[44], and validated on the other corpus. For evaluation, we used the Intersection over Union (IOU) metric, which is an upper bound of Aggregated Jaccard Index (AJI) metric[45], with a threshold of 0.5 to determine whether a detected cell is valid or not (Fig. 4f–j). We found that the Faster RCNN's performance on the combination of DAPI and pan-histone markers is better (AUC = 0.97) than either channel by itself (AUC = 0.88 for DAPI and 0.80 for pan-histone). To achieve scalable processing of large images with available GPU memory, we utilized the overlap-window strategy[46]. The end result is a set of identification tags (ID), locations, and bounding boxes for each nucleus.

**Identifying major cell types using Capsule Network**. The type, and when appropriate, the sub-type of each detected cell, was identified by analyzing the association of molecular biomarkers within their bounding boxes[47]. Since the signals are already corrected using computational synthesis of multiplex signal-corrected mosaic, they are directly summed over each box to quantify the overall expression level of each protein of interest. These data are exported to the cytometry tool FCS Express[48,49] for visualization. For identifying cell types, we used Capsule Networks[19,20,50,51], due to their ability to learn the object-part relationships of cells, built-in understanding of the 3D space, ability to learn from smaller training sets compared to CNNs, and their open-set classification ability[52] that allows us to assign ambiguous cells to the "unknown" type. However, the original Capsule Network is computationally expensive, motivating us to adopt the Fast CapsNet instead[53] (Fig. 5a). For generating the training sets in an automated manner, we used the 5,000 cells for each cell type with the highest expression level of the corresponding marker, e.g., NeuN for neurons, S100β for astrocytes, IBA1 for microglia, Olig2 oligodendrocytes, and RECA1 for endothelial cells.

Unlike a CNN whose output only indicates a cell as belonging to one of $K$ fixed classes, the CapsNet can also indicate unknown classes. It generates an array of 16-dimensional vectors for the known classes $\left\{\overrightarrow{V_1}, \overrightarrow{V_2}, \ldots, \overrightarrow{V_K}\right\}$. The entries of each output vector (i.e., its direction) represent the pose information (such as location, size, and orientation) of the associated cell type, while the length (the norm or magnitude) of the vector encodes the cell's existence probability. The vectors are normalized, with Euclidean lengths between 0 and 1. We used margin loss[20] to ensure that the lengths of these vectors are large if and only if a cell of the corresponding class exists in the image, and vice versa. If the lengths of all vectors are below a set threshold (0.5), the cell is assigned to the "unknown class". The histogram of lengths of capsule vectors exhibits a bimodal distribution for biomarkers that are present in the multiplex image (Fig. 5b). The major brain cell types can be identified reliably using this strategy (Fig. 5c). The deep network learns abstract representations of the input image (biomarkers of interest) and encapsulate the information into a vector in the last layer. The length of this vector contains comprehensive information (Fig. 5d–h right) compared to the traditional thresholding the histogram of mean intensity of the biomarker (Fig. 5d–h left). Figure 5d–h shows the ability of the proposed model in correcting the mis-identified cell types in 5 major cell types.

**Expanding cell phenotyping using morphological masking**. For identifying cell sub-types and phenotypic states, we found that the majority of phenotypic classifications follow a Boolean logic since most proteins of interest are either expressed strongly, or hardly at all, with a bi-modal histogram. For these cases, we introduced compound morphological masking based on combination of single or multiple channels consisting of different subcellular compartments (i.e., nucleus, soma, cytoplasm, plasma membrane, and cell processes). These masks were used to calculate the averaged marker expressions within each channel in order to threshold and assess whether the corresponding cell is either positive (+) or negative (−) for the molecular marker of interest. The cell type identifications and bounding boxes provided the basis for analyzing cell and arbor morphologies to the extent that they are revealed in the 10 µm thick slices to generate the compound morphological masks. The cell type information is used to select the molecular markers that indicate the major compartments of each cell. Given the sheer size of our images, we performed a simplified arbor analysis compared to our prior work[54–56]. For segmenting cell nuclei and soma, we used a Mask RCNN[57], and matched the results to the cell detections. We estimated cytoplasmic compartments by subtracting the nuclear masks from soma masks. We used skeletonization (62) to extract the basal processes, and directional ratios (63) to segregate soma from processes (e.g., IBA1 marks soma and processes for microglia, S100β marks soma and extends into processes for astrocytes). The directional ratios range from 0 to 1 with low values on processes and high values on soma. We delineated membranes by a 1-pixel dilation of the combined mask of soma, nucleus, and processes, in the absence of a membrane marker channel. For masking endothelial cells, we used RECA1 which marks plasma membranes of these cells as a validated membrane mask to trace endothelial cells. RECA1 was combined with endogenous GFP expression (a cytoplasmic label of brain endothelial cells in Lewis LEW-Tg (CAG-EGFP) YsRrrc transgenic rats) for endothelial cell reconstruction. Astrocytes were reconstructed using Sox2 (nuclear mask), S100β (soma mask), and GFAP (processes mask) channels. IBA1 channel was used to reconstruct microglia. MAP2 and NeuN channels were used for basal neuron arbor reconstruction. Olig2 (nuclear mask) and CNPase (soma and processes mask) were used to reconstruct oligodendrocytes. The morphological masking of subcellular compartments for each major cell type is illustrated in Fig. 6a–f. The generated masks were then used to calculate the average abundance of biomarker expression in each channel to expand the cell phenotyping. The resulting cell composition summary for the sample brain section is presented in Fig. 6g. As an example, we identified different neuronal cell subtypes using the proposed method based on transmitter phenotyping (i.e., Glutamatergic, GABAergic, Cholinergic, Catecholaminergic neurons), as shown in Fig. 6h–m.

Using the above steps, we generated a comprehensive output table containing the location, morphology, and phenotypic data of all cells (Supplementary Table 7) and a set of nucleus, soma, cytoplasm, and process morphological masks. Supplementary Table 7 includes data for 219,643 detected cells with phenotypic information of cell type, cell sub-type, and functional status from 28 unique biomarkers. These data can be profiled without restriction to generate readouts of the brain tissue at any scale. For example, these data can be parcellated using fitted brain Atlases. We fitted the Paxinos atlas to the corresponding section based on the coordinates using Adobe Photoshop (Fig. 6n). To quantify regional biomarker expression from Supplementary Table 7, we plotted the heatmap of the number of positive cells per $10^6$ pixel$^2$ area for the neurons (Fig. 6o) and the oligodendrocyte (Fig. 6p) populations.

**Reporting summary**. Further information on research design is available in the Nature Research Reporting Summary linked to this article.

## Data availability

The datasets including the original acquired images from the microscope and the results of each step are publicly hosted at https://doi.org/10.6084/m9.figshare.13731585.v1[58].

The cell nuclei detection model was trained on top of pre-trained "faster_rcnn_inception_resnet_v2_atrous_coco" model on ImageNet dataset. The pre-

trained weights can be found on https://github.com/tensorflow/models/blob/master/research/object_detection/g3doc/tf1_detection_zoo.md. The authors declare that all other data supporting the findings of this study are available within the paper and its supplementary information files.

## Code availability

Codes for reconstruction and quantitative analysis of the images are available at https://github.com/RoysamLab/whole_brain_analysis.git[28].

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

## Acknowledgements

This work was supported by NIH grant R01NS109118 and the Intramural Research Program of the National Institute of Neurological Disorders and Stroke, National Institutes of Health, and by the Mission Connect Foundation (Houston, TX). The authors wish to thank Dr. Prashant Prabhat from Semrock and Peter Brunt from AVR Optics for performing filter crosstalk analyses in SearchLight Spectra Viewer (https://searchlight.semrock.com/) to assess optimal custom filter set recommendations for 10-color epifluorescence imaging with minimal signal bleed-through. We would like to thank Dr. Sebastian Berisha and Dr. Saeed Ahmadian for their thoughtful comments and discussions.

## Author contributions

D.M. designed the multiplex IHC protocols and performed the multispectral imaging. J.J. developed a computational pipeline, fluorescence signal correction, and cell nuclei detection. X.R.L. developed registration and nuclei segmentation. A.M. and H.V.N. developed classification and J.J. modified for major cell type classification. Aditi Singh developed morphological masking. Andrea Sedlock performed brain tissue processing and multiplex IHC staining. K.G. developed an early prototype of the current system. B.R. guided the image analysis development. D.M., J.J., and B.R. wrote the manuscript and all authors commented on the manuscript.

## Competing interests

The authors declare no competing interests.
