## [Peer Review File · Nature Communications]

Reviewers' Comments:

Reviewer #1:

Remarks to the Author:

In this report, Maric et al. developed a whole brain tissue phenotyping pipeline for processing highly multiplexed immunohistological (MP-IHC) image datasets acquired via a customized wide-field multispectral epifluorescence imaging platform in conjunction with integrated open source computational modules for image reconstruction, optimization and quantitative deep machine learning data analyses. This approach hopes to accelerate pre-clinical drug evaluation and system-level brain histology studies by simultaneously profiling multiple biological processes in their native anatomical context.

Overall, the methodology developed in this study has several advantages: (1) the 10-plex IHC immunostaining method reported is relatively straightforward for implementation and adoption by broader neuroscience community; (2) Using thin sections imaged with epifluorescent microscope to avoid complex process that may compromise antibody staining and imaging quality; (3) the entire pipeline and computational algorithms are presented as an open resource to be shared with the community. The pipeline for transforming 2D to 3D brain histology for comprehensive cellular profiling from single and serial slices of brain tissue is very useful for broad neuroscience community; and finally (4) application of deep learning methods in automated signal segmentation remains a compelling point of this study.

However, I have several significant issues with the manuscript as cited below:

1. In terms of novelty, to my view, the experimental pipeline reported in this study including immunostaining, visualizing, reconstructing and analyzing 10 micron sections per round to localize several antigens in brain region(s) of interest, is similar to Stephen Smith's array tomography (Micheva and Smith, 2007, Neuron). Please cite this paper and discuss how the current study is different or improved in comparison with array tomography. I hope the deep learning based signal segmentation is not the only novel point.
2. In their figures, the authors showed coronal sections mapped onto a brain atlas, which is an essential step for quantifications of signals in different brain structures. It appears that the authors tried to map coronal brain sections onto the Paxinos and Watson (PW; 2014) atlas based on Bregma/interaural coordinates but did not address how they reconciled and mapped sections that fall in between atlas levels (this is important, because their coronal sections are at 10um thickness, which is much thinner than the sections used in Paxinos and Watson atlas). The authors also did not explain how they 'manually fitted' the parcellations of the atlas onto their sections (Figure 6 n-p). Furthermore, if this step needs to be manual, it will be very tedious and labor intensive. Probably the authors can read a useful reference (Khan et al., 2018; Front Syst Neurosci., <https://pubmed.ncbi.nlm.nih.gov/29765309/>) for how to register rat brain images onto a standard brain atlas.
3. The authors mentioned and depicted in their figures about 3D reconstruction from MP-IHC image datasets of serial whole rat brain tissue sections imaged in 2D. However, description of this step is missing from their Results section.
4. Cryostat cutting of thin sections tends to cause drastic non-uniform deformation, which makes it difficult to register and render serial sections into 3D. The authors did not sufficiently describe how they solved this issue. Also, serial sections at 10 μm thickness for a whole rat brain is excessive and does not seem practical to render all these sections together into a 3D brain. Additionally, confocal microscopes are routinely used in most neuroscience labs, which are suitable to image much thicker sections (e.g., 200μm). Is this pipeline amenable for thicker sections imaged with confocal and other microscopes?

5. About "Signal is more robust than background". They did not discuss the random bright spots/debris that can appear in the background. This could artificially inflate the analysis. Autofluorescence correction methods are unclear.

Reviewer #2:

Remarks to the Author:

The authors provide a protocol for sample preparation, imaging, and analysis of phenotypic spatial data for multispectral images. They have scaled up multispectral imaging with 10 excitation/emission filters and a cocktail of 10 antibodies and 10 more IGg. Large scale mosaic images are generated and samples are registered at each iteration. The method is integration of established technology and should be of interest to a wider community. Their goal is to count and delineate each cell in the tissue, measure its morphometry and its protein expression so that standard statistical methods can be applied for population studies.

Minor issues: In the revision, please indicate

- 1) The mouse antibody stain samples from rat, but will this cocktail limit non-specific binding as designed?
- 2) Performance of nuclear segmentation is often reported at the AJI score. You may want to use similar convention.
- 3) It will be great to show "a" visualization of 10 channel readouts by one of the flow cytometry code.
- 4) Since phenotypic readout also preserve spatial relationships, how do you anticipate to leverage organization information?

Reviewer #3:

Remarks to the Author:

The manuscript entitled "Comprehensive cell phenotyping method for whole-brain tissue mapping using highly multiplexed immunofluorescence imaging, computational reconstruction, and deep neural networks" touches upon an important topic of multiplex profiling of cell status in situ. The methods developed for the pipeline are sound and easy to implement by regular labs. The topic, significance and novelty of this manuscript fit well for Nature Communications. I encourage to accept this manuscript after addressing the concerns listed below.

Major points

- 1) As 2D images are taken from 10 micron thick sections, it is possible that the same nucleus is identified from adjacent sections but potentially being assigned as distinct cell type due to the non-uniformly distributed molecular markers or cell morphology. How is this error being estimated and corrected?
- 2) It will be beneficial to show processing of the same region in adjacent sections and indicate how tissue loss is estimated. Addressing how this error may potentially be corrected as an alternative to methods that rely on whole brain 3D imaging is helpful to broaden the potential impact of the presented method. For instance, compare the number of all neurons or other cell types counted throughout the whole brain.
- 3) The 3D volume reconstruction was also mentioned in the last sentence of the Results section: "Finally, we expanded the analysis of the proposed pipeline (Fig. 1a) for processing 3D brain immunohistology datasets by combining the reconstructed 2D results from the serial whole-brain sections to derive a volumetric dataset in context of 3D brain anatomical mapping (e.g., Fig. 1b)". However, the example provided in Fig 1b is a simple 2D stacking. True 3D volume presentation of sub brain regions, such as CA1, is needed to make this statement sound.

4) The establishment of the Capsule Network needs more detailed information. For instance, pointing out what exactly the vectors are quantified for in "It generates an array of 16-dimensional vectors for the known classes...". This helps broader readers to understand the model and potentially reduce the difficulty of using the tool. When comparing Fig 5a to 5c, it is better to show some cell image examples of how mis-identified cell type in 5a being correctly identified in 5c.

5) While it is nice to show the enumeration of cell types and quantification statistics for the whole section in Fig 6g, it is important to show data from brain regions, such as somatosensory cortex and visual cortex, in which cell types are well documented by other methods such as scRNAseq. A brief comparison between different methods is helpful to determine the faithful representation of the brain cell type distributions.

Minor points

1) Dye spectra shown in Fig 2a do not match the exact dyes being used in Fig 2b.

2) It is confusing that the authors wrote "The raw images are registered (Section OM.C) and stitched (Section OM.D) to correct for stage alignment errors using the microscope's Zen software (Zeiss) and for computing an affine spatial transformation for pixel-to-pixel registration (Fig. 3a)", which indicates that each round of images are "online" (?) stitched to a full section mosaic by Zen first, but in OM.D the authors wrote "To achieve scalable handling of mosaic images, we implemented RANSAC over image tiles rather than the full mosaic".

3) Fig 5a has been only referenced in OM.G, which should be introduced in the main text before Fig 5b.

4) Numbers in Fig 6g do not add up to proper values. For instance, the total number of the four listed neuronal subtypes are $67,703 + 2,849 + 94 + 128 = 70,774$; and the sum of percentages of all cell types is not 100%.

5) Replace Fig 6i with four smaller panels to show seed detections for each neuronal subtype.

6) Fig 6o, 6p: not sure what the numbers 2754 and 2302 stand for.

AUTHOR RESPONSE TO REVIEW COMMENTS

The authors wish to thank the reviewers for the kind comments and helpful suggestions. We have worked to answer each of the queries and incorporate each of the suggested improvements to the manuscripts. The following paragraphs detail our actions.

Reviewer #1 (Remarks to the Author):

In this report, Maric et al. developed a whole brain tissue phenotyping pipeline for processing highly multiplexed immune-histological (MP-IHC) image datasets acquired via a customized wide-field multispectral epifluorescence imaging platform in conjunction with integrated open source computational modules for image reconstruction, optimization and quantitative deep machine learning data analyses. This approach hopes to accelerate pre-clinical drug evaluation and system-level brain histology studies by simultaneously profiling multiple biological processes in their native anatomical context.

Overall, the methodology developed in this study has several advantages: (1) the 10-plex IHC immunostaining method reported is relatively straightforward for implementation and adoption by broader neuroscience community; (2) Using thin sections imaged with epifluorescent microscope to avoid complex process that may compromise antibody staining and imaging quality; (3) the entire pipeline and computational algorithms are presented as an open resource to be shared with the community. The pipeline for transforming 2D to 3D brain histology for comprehensive cellular profiling from single and serial slices of brain tissue is very useful for broad neuroscience community; and finally (4) application of deep learning methods in automated signal segmentation remains a compelling point of this study.

Author response: We thank the reviewer for the kind comments.

However, I have several significant issues with the manuscript as cited below:

1. In terms of novelty, to my view, the experimental pipeline reported in this study including immunostaining, visualizing, reconstructing and analyzing 10 micron sections per round to localize several antigens in brain region(s) of interest, is similar to Stephen Smith's array tomography (Micheva and Smith, 2007, Neuron). Please cite this paper and discuss how the current study is different or improved in comparison with array tomography. I hope the deep learning based signal segmentation is not the only novel point.

Author Response: We thank the reviewer for this comment, and we have added a citation to the pioneering Micheva and Smith 2007 array tomography paper.

Aside from the large-scale automated quantitation, our work represents a different set of tradeoffs in terms of tissue imaging, in order to serve a different operational objective - provide a practical fully integrated and readily accessible method that can be used in ordinary laboratories for accelerating drug discovery efforts by providing high levels of

multiplexing, wide region coverage (whole-brain slices), and multi-scale automated quantitation.

First, the difference in emphasis in our method compared to that of Micheva and Smith entails a different tradeoff to high spatial resolution. While the array tomography method by Micheva and Smith broke new ground in terms of imaging at high spatial resolution (~50 nm/pixel, 3-D) by using a specialized tissue slicing technique similar to electron microscopy, the overall extent of brain tissue imaging reported in this study covered only about 1mm of specimen thickness, making this method more suitable for high-resolution studies of small brain. In contrast, our method uses thicker slices (10 μm) sourced by a conventional tissue slicing technique, which is widely available in ordinary laboratories and core facilities, and works at a lower lateral imaging resolution of 325 nm/pixel, which is sufficient for cellular-resolution quantitation, but yielding a much larger tissue extent (whole-brain slices). This is important since cellular alterations in the brain, particularly after brain injury, are often linked to changes across wide areas of the brain that may be distant from the immediate sites of injury, thus requiring large-scale tissue imaging to capture these changes. This advance is enabled by our computational image alignment and signal correction methods.

Furthermore, Micheva and Smith study used only a 3-plex biomarker screening protocol at each staining/imaging round, and screened ~30 total biomarkers, mostly focused on neuronal cell phenotyping. In contrast, our method incorporates a 10-plex screening protocol at each staining round, thus allowing for accumulation of many more markers, with a much longer “stride” of 10 compared to a 3-plex protocol. This allows us to capture a broad spectrum of biomarker panels designed to phenotype all cell classes in brain tissue (neuronal, glial, and vascular), to enable a systems approach. Our primary goal is to identify, delineate, classify and anatomically map all the major resident cell phenotypes (i.e., neurons and specific neuronal subtypes, astrocytes, oligodendrocytes, neural stem cells, ependymal cells, microglia, and blood vessel cell types including endothelial cells and pericytes) in whole brain tissue sections, thus allowing for a more comprehensive systems biology approach to study brain cytoarchitecture and function at multiple scales of spatial resolution from individual cells to whole anatomical regions. From an application perspective, for example, drug evaluation studies, our method can be used as a comprehensive assay that eliminates the need to painstakingly piece together information from multiple low-content assays, and provides full spatial context at the cellular scale.

Taken together, our method thus provides a more efficient and higher-content histological biomarker screening, comprehensive cellular mapping to assay of whole brain systems biology on multi-scale levels, with widely available accessibility to ordinary laboratories and conventional core facilities.

2. In their figures, the authors showed coronal sections mapped onto a brain atlas, which is an essential step for quantifications of signals in different brain structures. It appears that the authors tried to map coronal brain sections onto the Paxinos and Watson (PW; 2014) atlas based on Bregma/interaural coordinates but did not address how they reconciled and mapped sections that fall in between atlas levels (this is important, because their coronal

sections are at 10µm thickness, which is much thinner than the sections used in Paxinos and Watson atlas). The authors also did not explain how they ‘manually fitted’ the parcellations of the atlas onto their sections (Figure 6 n–p). Furthermore, if this step needs to be manual, it will be very tedious and labor intensive. Probably the authors can read a useful reference (Khan et al., 2018; Front Syst Neurosci., <https://pubmed.ncbi.nlm.nih.gov/29765309/>) for how to register rat brain images onto a standard brain atlas.

Author Response: We thank the reviewer for this comment. We have now added the Khan et al. reference into the manuscript, with an explanation. Basically, our goal here is to illustrate logical ways to parcellate the cellular-scale measurement data, and atlas fitting is clearly one such approach. However, the atlas fitting performed here is necessarily approximate compared to what is reported in the literature due to the limitations in the precision of slice positioning relative to the Bregma, and the higher slice thickness compared to Dr. Paxinos’ studies.

However, our results make clear that, even with the approximate atlas fitting, our method yields a bountiful and informative way to profile the cytometric data by brain region. We agree with the reviewer’s guidance that a more accurate atlas fitting can only make these results more accurate, and we will adopt and develop more accurate atlas fitting methods in the future.

The Paxinos atlas is not the only way to parcellate the cellular data – the user can define any regions of interest), for example, sites of brain injury, using common image annotation tools (e.g., Photoshop) in order to parcellate the data.

3. The authors mentioned and depicted in their figures about 3D reconstruction from MP-IHC image datasets of serial whole rat brain tissue sections imaged in 2D. However, description of this step is missing from their Results section.

Author Response: We have added a brief description of our method, and further clarified our purpose. Our 3D reconstruction has the limited goal of expanding the two-dimensional (2D) immuno-histology results to 3D volumetric datasets by associating and combining the cellular phenotyping across adjacent thin serial sections. This provides a way to sample regions that are thicker than what is possible by the 10 µm sectioning, while also serving to confirm the consistency of the cellular distribution in the neighboring sections. However, given the optical resolution of ~325 nm/pixel, our 3D alignment is clearly not precise enough for connectomics analysis, for example.

4. Cryostat cutting of thin sections tends to cause drastic non-uniform deformation, which makes it difficult to register and render serial sections into 3D. The authors did not sufficiently describe how they solved this issue. Also, serial sections at 10 µm thickness for a whole rat brain is excessive and does not seem practical to render all these sections together into a 3D brain. Additionally, confocal microscopes are routinely used in most neuroscience labs, which are suitable to image much thicker sections (e.g., 200µm). Is this

pipeline amenable for thicker sections imaged with confocal and other microscopes?

Author Response: We have added an explanation to the manuscript. We agree with the reviewer that 10 μ m slices are too thick if high-resolution 3D imaging of the tissue was the primary objective. As noted above, this is not our goal at all. The primary purpose of our method is to identify the detailed phenotypic status of each cell in individual tissue sections, and secondarily, to enable a limited form volumetric sampling across multiple serial sections when needed. With this in mind, the 10 μ m slice thickness represents an ideal tradeoff since the vast majority of drug-discovery related studies can be carried out with 10 μ m slices, allowing us to assess brain system biology on multiple scales of spatial resolution ranging from subcellular structures, individual cells, multicellular niches, and anatomical regions of interest.

The 10 μ m thick serial sections are preferable to thicker sections (>20 microns thick) since they allow uniform immunohistochemical (IHC) staining for all tissue biomarkers of interest. In our experience, sections thicker than 10 μ m do not provide uniform antibody staining results under gentle iterative antigen retrieval conditions used in our pipeline, and need to be permeabilized using harsh detergents (e.g., Triton X-100), which compromise membrane associated and lipid soluble antigens and thus reduce the reliability of cell phenotyping using antibodies targeting these antigens.

Our pipeline is indeed amenable to confocal imaging should the need arise. For example, the immunolabeled 10 μ m thin sections can be subjected to confocal imaging for capturing high-content information on individual cells, or localized regions of interest, at high spatial resolution using high numerical aperture (NA) 60X and 100X objectives compared to the 20X/0.8 NA objective used in our multispectral epifluorescence imaging system.

It must be noted that most contemporary confocal systems are not equipped for 10-color fluorescence imaging, and the highly customized ones that can image 10 fluorophores are complex and prohibitively expensive for individual labs, whereas the 10-color epifluorescence microscopes used here are straightforward and affordable for widespread adaptation of our pipeline.

5. About "Signal is more robust than background". They did not discuss the random bright spots/debris that can appear in the background. This could artificially inflate the analysis. Autofluorescence correction methods are unclear.

Author Response: Since the random bright spots/debris are much larger than the structure of the interest in each channel, the intra-channel correction filters usually take care of the signal from debris. In case that the imaging artifact is not corrected in the intra-channel fluorescence correction step, we proposed an extension to the inter-channel correction module to use a Region Of Interest (ROI) where the majority of the signal relies for two purposes. First, using an ROI helps estimating the unmixing parameters in a smaller dimension space which makes the algorithm faster. Second, the ROI, specified by user, is guaranteed to have the brightest signal from the structure of interest and it is debris-free. We can threshold any pixel that is brighter than the brightest pixel in the ROI to remove the

artifacts. A short paragraph addressing the bright spots/debris correction has been added to the manuscript in Online Methods section E.

Reviewer #2 (Remarks to the Author):

The authors provide a protocol for sample preparation, imaging, and analysis of phenotypic spatial data for multispectral images. They have scaled up multispectral imaging with 10 excitation/emission filters and a cocktail of 10 antibodies and 10 more IgG. Large scale mosaic images are generated and samples are registered at each iteration. The method is integration of established technology and should be of interest to a wider community. Their goal is to count and delineate each cell in the tissue, measure its morphometry and its protein expression so that standard statistical methods can be applied for population studies.

Author Response: We thank the reviewer for the kind comments, and we certainly hope to serve the wider community by this carefully designed approach.

Minor issues: In the revision, please indicate

1) The mouse antibody stain samples from rat, but will this cocktail limit non-specific binding as designed?

Author Response: This concern has been addressed with added explanation in the methods section. The antibodies which we included in the cocktail panels for multiplexed IHC staining are well-known in the literature and have been individually validated for the rat tissues without non-specific binding issues. In order to ensure the immunocompatibility of the multiplex antibody panels, we combined antibodies from different hosts and/or different immunoglobulin classes/subclasses to construct highly compatible multiplex IHC antibody panels that did not show significant cross-reactivity nor non-specific binding. All antibodies have been individually validated in our lab to be immuno-compatible for multiplexing as antibody cocktails that do not cross-react with each other. Furthermore, all tissues specimens were pre-blocked against potential nonspecific antibody binding or cross-reactivity using a combination of FcR and background blocking solutions prior to applying primary and secondary antibody cocktails in each immunostaining round.

2) Performance of nuclear segmentation is often reported at the AJI score. You may want to use similar convention.

Author Response: In the revised manuscript, we have added an explanation. The IoU measure that we are reporting is the upper bound of the AJI score that is appropriate for our study, as explained below. In this manuscript, we have only reported the bounding box detection IoU performance by Faster RCNN, since: (i) the majority of the cell phenotyping can be performed reliably using the bounding boxes alone; and (ii) manual ground truth data for cell detection (bounding boxes) is currently practical to generate, and we had these data available.

However, generating full segmentation ground truth for nuclear segmentation at the scale of this study, where each image can contain ~300,000 cell nuclei, many of which can overlap given the 10 μm slice thickness, is a dramatically more ambitious task, and well outside the scope of the present study. We plan to develop and report sampling-based methods to quantify nuclear segmentation accuracy using pixel-level scores in the future.

3) It will be great to show “a” visualization of 10 channel readouts by one of the flow cytometry code.

Author Response: This has been done. A set of images visualizing the original images and overlaid results using flow cytometry is now added to revised Figure 5 to both visualize the readouts and illustrate the power or proposed deep learning classification method in outperforming the existing phenotyping methods based solely on signal intensity thresholding.

4) Since phenotypic readout also preserve spatial relationships, how do you anticipate to leverage organization information?

Author Response: We have updated the manuscript explaining some ways we can leverage the organizational information both in “Deep learning-based multiplex quantification” section of the Results and the Discussion chapters. This is a surprisingly rich topic, since the organization occurs at multiple spatial scales, and much remains to be explored. We elaborated that the phenotypic readouts can be grouped based on the anatomical regions which gives us the leverage to study the differences in the population of cells in different anatomical regions or between same regions in different sections or different animals.

Reviewer #3 (Remarks to the Author):

The manuscript entitled “Comprehensive cell phenotyping method for whole-brain tissue mapping using highly multiplexed immunofluorescence imaging, computational reconstruction, and deep neural networks” touches upon an important topic of multiplex profiling of cell status in situ. The methods developed for the pipeline are sound and easy to implement by regular labs. The topic, significance and novelty of this manuscript fit well for Nature Communications. I encourage to accept this manuscript after addressing the concerns listed below.

Author Response: We thank the reviewer for the kind endorsement of the importance of this topic, the practicality of our approach, and the novelty of our work.

Major points

1) As 2D images are taken from 10 micron thick sections, it is possible that the same nucleus is identified from adjacent sections but potentially being assigned as distinct cell type due to the non-uniformly distributed molecular markers or cell morphology. How is this error being estimated and corrected?

Author Response: Yes, it is indeed possible for the tissue slicing to separate a cell nucleus between adjacent tissue slices, but this error was not estimated in this study for the following reasons. From a cell-phenotypic profiling standpoint, these errors have no impact on the study of individual slices, which is the most common use case for the proposed method. They only affect cell counts in studies that analyze stacks of adjacent serial sections since the split cells are counted twice, and even there, these effects do not have a bias as to cell type.

2) It will be beneficial to show processing of the same region in adjacent sections and indicate how tissue loss is estimated. Addressing how this error may potentially be corrected as an alternative to methods that rely on whole brain 3D imaging is helpful to broaden the potential impact of the presented method. For instance, compare the number of all neurons or other cell types counted throughout the whole brain.

Author Response: This is clearly a very desirable goal, but it would be well beyond the scope of the present study. The primary goal of our study was to phenotype and enumerate cells in individual sections in 2D slices and reconstruct volumetric data across multiple adjacent serial sections whenever it becomes important from a brain tissue assay standpoint, rather than to precisely align sections or reconstruct cell connectivity in 3D.

3) The 3D volume reconstruction was also mentioned in the last sentence of the Results section: “Finally, we expanded the analysis of the proposed pipeline (Fig. 1a) for processing 3D brain immunohistology datasets by **stacking** the reconstructed 2D results from the serial whole-brain sections to derive a volumetric dataset in context of 3D brain anatomical mapping (e.g., Fig. 1b)”. However, the example provided in Fig 1b is a simple 2D stacking. True 3D volume presentation of sub brain regions, such as CA1, is needed to make this statement sound.

Author Response: The reviewer’s understanding is accurate. We have reworded the manuscript to make this statement clearer. Our aim was not to present a method for true 3D reconstruction representing the connectome of brain cytoarchitecture. We have removed the word “reconstruction” in the revised manuscript since we are using the 2D immunohistology results from individual slices and stacking them, to borrow the reviewer’s suggestion, by associating the cellular phenotyping across adjacent thin serial sections. We thus combined the results from multiple 2D sections into a single graph to illustrate the consistency of the cellular distribution in the nearest-neighbor sections.

4) The establishment of the Capsule Network needs more detailed information. For instance, pointing out what exactly the vectors are quantified for in “It generates an array of 16-dimensional vectors for the known classes...”. This helps broader readers to understand the model and potentially reduce the difficulty of using the tool. When comparing Fig 5a to 5c, it is better to show some cell image examples of how mis-identified cell type in 5a being correctly identified in 5c.

Author Response: We thank the reviewer for pointing out the missing information. A paragraph describing the Capsule Network has been added to the section G of the Online Methods to provide more details. Also, a set of images visualizing the original images and overlaid results using flow cytometry is now added to Figure 5, both to visualize the readouts, and to illustrate the power of the proposed classification method.

5) While it is nice to show the enumeration of cell types and quantification statistics for the whole section in Fig 6g, it is important to show data from brain regions, such as somatosensory cortex and visual cortex, in which cell types are well documented by other methods such as scRNAseq. A brief comparison between different methods is helpful to determine the faithful representation of the brain cell type distributions.

Author Response: We agree with the reviewer that this would be a valuable additional comparison between IHC results and other methods such as scRNAseq. However, in the present study, we used antibodies that are well-characterized and were validated extensively in our lab to be consistent with the observations in the context of the background literature, so we are confident that our image-based IHC measurements provide a sufficiently faithful representation of the brain cell types for laboratory studies.

Minor points

1) Dye spectra shown in Fig 2a do not match the exact dyes being used in Fig 2b.

Author Response: We have now removed Figure 2a to avoid this confusion. Thank you.

2) It is confusing that the authors wrote “The raw images are registered (Section OM.C) and stitched (Section OM.D) to correct for stage alignment errors using the microscope’s Zen software (Zeiss) and for computing an affine spatial transformation for pixel-to-pixel registration (Fig. 3a)”, which indicates that each round of images are “online” (?) stitched to a full section mosaic by Zen first, but in OM.D the authors wrote “To achieve scalable handling of mosaic images, we implemented RANSAC over image tiles rather than the full mosaic”.

Author Response: This passage has been reworded to eliminate the confusion for which we apologize. The images collected within each staining round are stitched using the built-in Zen software. Next, we used a registration algorithm based on RANSAC to correct for alignment errors arising between staining rounds. For this, we use the full mosaic reconstructed in the previous step using the Zen software. But we use a scanning window (tiles) to extract the keypoints in a computationally parallel manner for increasing the speed of analysis. These latter tiles have nothing to do with the original tiles (microscope fields) and are solely used to speed up the registration algorithm. The manuscript is updated accordingly.

3) Fig 5a has been only referenced in OM.G, which should be introduced in the main text before Fig 5b.

Author Response: We thank the reviewer for catching this editing oversight. The manuscript has been updated to address this error.

4) Numbers in Fig 6g do not add up to proper values. For instance, the total number of the four listed neuronal subtypes are $67,703 + 2,849 + 94 + 128 = 70,774$; and the sum of percentages of all cell types is not 100%.

Author Response: These numbers have now been corrected in the updated manuscript. The reason for the numbers not adding up is that the uncharacterized neurons are not reported in the table. This population is now added to make the table clearer.

5) Replace Fig 6i with four smaller panels to show seed detections for each neuronal subtype.

Author Response: This is already the case. The smaller panels of detected and classified seeds of neuronal subtypes are presented in Fig 6 j-m

6) Fig 6o, 6p: not sure what the numbers 2754 and 2302 stand for.

Author Response: These numbers represent the cell counts. We have updated the figure caption to make this clearer.

Reviewers' Comments:

Reviewer #1:

Remarks to the Author:

The authors have addressed all of my critiques. No more comments from this reviewer.

Reviewer #2:

None

REVIEWERS' COMMENTS

Reviewer #1 (Remarks to the Author):

The authors have addressed all of my critiques. No more comments from this reviewer.

Authors response: we thank the reviewer for providing the professional review.

Reviewer #1's response to the rebuttal of Reviewer #3's comments:

Major concerns

1) The authors' response is satisfactory. In the rebuttal, the authors detail very clearly how they minimized the error of identifying one neuron/nucleus multiple times. I suggest they include this rationale in the discussion; relevant references will further bolster their claim.

Authors response: we thank the reviewer for the kind endorsement. We have updated the discussion clarifying that our work does not seek to identify cells across successor sections, and we have added the relevant references.

2) The authors' response is satisfactory. It would be beneficial to mention the present proof-of-concept studies may be applied to increase the rigor of future studies.

Authors response: we thank the reviewer for the kind endorsement. We have added a sentence that our 2D cell analyses provides an additional tool for rigorous scalable validation of future 3D analyses.

3) The reviewer is critical of the authors' use of 3D reconstruction, which authors have thoughtfully changed in the narrative.

Authors response: we apologize for the ambiguity regarding the word "reconstruction" and we reworded the manuscript to address the ambiguity.

The authors have addressed all of other concerns from Reviewer #3. No more comments from myself.